# 50 GHz Four-Port Coupling-Reduced Probe Card Utilizing Pogo Pins Housed in Custom Metallic Socket

**DOI:** 10.3390/s24123745

**Published:** 2024-06-09

**Authors:** K. M. Lee, J. S. Kim, S. Ahn, E. Park, J. Myeong, M. Kim

**Affiliations:** 1School of Electrical Engineering, Korea University, Seoul 02841, Republic of Korea; leemin7@korea.ac.kr (K.M.L.);; 2Samsung Electronics Co., Ltd., Suwon 16677, Republic of Korea; sanguck.ahn@samsung.com (S.A.);

**Keywords:** RF probe cards, mm-wave multi-port testing, metallic socket, coupling removal technique

## Abstract

A design for a pogo-pin probe card featuring a metallic socket is proposed to eliminate signal leakage and coupling loss in a multi-port environment. The proposed metallic pogo-pin socket includes a metal wall structure between adjacent pogo pins, ensuring complete isolation. This metal wall offers an advantage in removing coupling issues between pogo pins that can occur with typical dielectric pogo-pin sockets. The designed probe card is fabricated as a prototype and verified for its performance. Measurement results using a test through line show that coupled power is minimized, providing a low-loss transmission performance of −2.14 dB to an RF chip at 50 GHz, all within a compact size. Although the dielectric spacer used to secure the pogo pins allows for some leakage, it can maintain a low coupling performance of under −15 dB in the millimeter-wave band. The prototype probe card can deliver an RF signal to a 5G circuit with a low loss of −0.7 dB at 28 GHz and −1.9 dB at 39 GHz frequency. The designed probe card is capable of transmitting multiple RF signals to the RF system without signal distortion in a multi-port environment.

## 1. Introduction

As research on high-frequency band RF systems over 20 GHz, such as 5G systems, increases [1], interest in multi-port chip measurement technology is also increasing. In the high-frequency band, multi-port measurement technology is important in the on-wafer test stage because high-frequency integrated chips are often designed and fabricated as multi-input multi-output (MIMO) arrays due to the low gain and low output power of individual antennas and RF devices [2]. To measure these multi-port systems in an on-wafer environment, a system capable of measuring MIMO chips is being studied and manufactured in the high-frequency band [3]. These systems should include an RF probe structure capable of multi-port measurements. Typically, an RF probe card equipped with tips fabricated using the MEMS process is used for on-wafer RF chip measurements [4]. This type of commercialized RF probe card product is fabricated by companies such as GGB and T-plus up to high-frequency bands [5,6]. However, commercialized multi-port probe cards are limited by a simple arrangement of single probes, restricting the structural shape and the number of array tips [7]. Furthermore, the durability of the probe tip is often low, and the probe tip is easily worn with overdrive [8,9]. To address these issues, the pogo-pin probe card, which offers high durability with spring compression and the convenience of arrangement, has been proposed for replacing the probe card with a probe tip.

Various studies are underway to explore the use of pogo pins as replacements for RF probes in low-frequency bands because pogo pins can be arranged freely, unlike typical probe tips. Probe cards utilizing pogo pins in low frequencies, such as MHz bands, were commonly fabricated and employed [10]. Additionally, there have been studies analyzing the characteristic impedance of pogo pins for use in several GHz bands, along with research on optimizing the pogo-pin structure [11]. In addition to structural analysis, a study has been conducted to analyze the insertion loss performance by fabricating actual pogo pins and sockets up to the 10 GHz band [12]. Another study verified the insertion loss of pogo pins up to a 20 GHz frequency by using an RF probe suitable for higher frequency bands than an SMA connector [13]. Several studies have demonstrated the performance of pogo pins in low-frequency bands below 20 GHz. However, in high-frequency bands such as millimeter waves, various issues arise that are not present in lower-frequency bands. The pogo-pin structure itself can be used in frequency bands above 20 GHz. Examples of pogo-pin cables demonstrate that they can transmit RF signals above 20 GHz [14,15]. A study on pogo-pin cables showed that signal transmission is feasible up to 40 GHz, which is within the millimeter-wave band, using a pogo-pin structure [16]. However, this particular structure features a wide spacing between pins of 25.4 mm, which differs from the system-on-chip (SoC) testing environment. Pasternack also offered a commercial single RF probe using pogo pins that guarantees performance up to 40 GHz [17]. These references demonstrate that pogo pins are suitable for use in the millimeter-wave band. However, both the pogo-pin RF probe and cable face a challenge with wide pin spacing, with a pitch exceeding 800 µm, making them unsuitable for measuring integrated chips (IC) in the on-wafer state.

Recent research is exploring pogo-pin sockets designed for use in millimeter-wave bands within SoC environments. One study indicated the potential for resonance occurring at 30 GHz due to the height of the dielectric socket, as demonstrated through simulations with a pogo-pin array [18]. This study highlighted challenges with pogo-pin arrays and dielectric sockets in the analysis of transmission and coupling performance. Another analysis involving a pogo-pin prototype was conducted to verify performance in the millimeter-wave band [19]. However, this study revealed challenges in the analysis due to high insertion loss and the probe card structure used. The probe card described in this reference makes it difficult to identify problems occurring in actual SoC tests because the prototype is fabricated with a vertical structure that differs from typical probe cards. To solve this issue, some studies have confirmed performance up to 50 GHz by fabricating a prototype with an environment similar to an actual probe card, using a combination of pogo pins and a dielectric socket [20]. This study explains that a probe card, including a dielectric socket and printed circuit board (PCB) for signal transmission, can act as a dielectric resonator satisfying a specific wavelength in the millimeter-wave band. The measurement results indicate that resonance occurs at 28 GHz, which is utilized in the 5G system. To address the resonance problem, a pogo-pin probe card incorporating resonance removal techniques has also been designed [21]. However, these studies are limited to only demonstrating performance in a single-port configuration. In single-port measurement environments, a pogo-pin probe card with a dielectric socket may show potential for replacing an RF probe in the millimeter-wave band. However, analyzing only in a single-port environment limits the ability to identify problems that may occur in setups with multiple ports.

Recent studies have also explored the benefits of using two or more pogo pins to measure arrays of pogo pins. One simulation analysis was conducted to verify the placement of ground pins that minimize coupling loss between pogo pins up to 10 GHz [22]. Additionally, another study involved measuring coupling loss up to the 10 GHz band using two fabricated pogo pins and analyzing the array of ground pins [23]. While studies conducted in low-frequency bands below 10 GHz show improvements in reducing coupling components, achieving the same effect at higher frequencies is challenging. Therefore, it is necessary to conduct a multi-port pogo-pin probe card analysis, specifically in the millimeter-wave band, to address these challenges. Because the dielectric socket relies solely on an isolation structure through a ground pin array to prevent signal leakage between signal pins, it is unable to completely eliminate the coupling component between signal pins. As the frequency increases, signal leakage also increases, and the problem of coupled power easily occurs in the millimeter-wave band. To solve this problem, changing the socket material is recommended as a solution. Studies have explored using a coaxial structure with pogo pins at low frequencies below 10 GHz [24]. Additionally, simulations have demonstrated a decrease in the coupling component up to 20 GHz when a metallic socket is used [25]. However, previous studies have shown a trend where mismatch increases as the frequency band rises, rendering designed probe cards unusable in the millimeter wave range and limiting verification to simulation results alone. Another challenge is the difficulty in confirming improvements in coupling loss performance in the high frequency, as these studies primarily focus on analysis limited to low-frequency bands.

This paper proposes a pogo-pin probe card using a metallic pogo-pin socket designed for use in the millimeter wave band. The problems of typical pogo-pin probe cards used in the high-frequency band are described in detail in Section 2. Section 3 proposes a probe card structure that can solve the problems described in Section 2. The proposed probe card is designed to be as compact as possible to satisfy the minimum insertion loss, and the main body of the socket has been replaced with a metal material. The socket structure, including a metal wall, is described in detail in Section 3. Using the metal wall effect, the proposed probe card can offer a complete isolation structure between signal pogo pins, thereby eliminating signal distortion caused by coupled power in multi-port environments in millimeter-wave frequencies. A spacer and a non-shorting gap structure are designed to prevent RF shorting, which is a problem that can occur in a metallic socket. The analysis of the coupling problem, the prototype fabrication information, and the measurement setup of the proposed probe card are included in Section 4. Section 5 validated the claims of this paper through the fabricated prototype probe card. The performance in removing coupling components in a multi-port environment is verified with measurements using a prototype probe card. Finally, Section 6 consists of a summary and technical applications with the proposed pogo-pin probe card.

## 2. Problem of Typical Dielectric Pogo-Pin Socket

The commonly used pogo-pin socket is typically made of a dielectric material, which offers advantages in terms of fabrication and ease of securing pogo pins. In a previous study, the pogo-pin socket was made of MDS100 material (dielectric constant 3.37, Leeno Industrial Inc., Busan, Korea), which is commonly used in socket fabrication due to its low-loss characteristics [21]. The basic structure of the previous socket consisted of a dielectric main body, including a pogo-pin array and a large dielectric holder that secured the socket main body. While this type of pogo-pin socket poses no issues at low frequencies, using it at higher frequencies, such as millimeter waves, presents two problems.

The first problem is related to insertion loss. Due to the size of the dielectric holder structure, the overall size of the dielectric pogo-pin socket is 44 mm in length, which is a large structure in the millimeter wave band. Including the dielectric permittivity, this length corresponds to 13.5 wavelengths (λ) at 50 GHz, potentially leading to increased insertion loss. Moreover, the large socket structure necessitates a correspondingly long PCB line for the probe card. Taking into account connector connections, the prototype PCB must be sized at 66 mm [21]. This length is a short distance in the low-frequency band, including DC, resulting in almost no insertion loss. However, in a high millimeter-wave band such as 50 GHz, the length becomes 20.1 λ, which can lead to an insertion loss problem. In the previous study [21], the length of the entire PCB line was 29 λ, demonstrating an insertion loss performance of 8 dB at 50 GHz. Such a high insertion loss problem can make it difficult to transmit signals to an RF chip and analyze issues caused by the dielectric socket structure due to high PCB dielectric loss. The second problem is the coupling issue between adjacent pogo pins. A common challenge in densely packed multi-port RF signal line environments, particularly in the gigahertz frequency band, is coupling between signal lines. In a previous study [20,21], RF pogo pins inside the dielectric socket were placed very closely with an interval of 0.522 mm. This distance, when expressed in terms of wavelength, indicates that the pogo pins were spaced approximately 0.096 λ at a frequency of 30 GHz. When an RF signal passes through two adjacent pogo pins, there is a risk of signal coupling and interference between the two lines. Because typical pogo-pin sockets are made of dielectric material, there is no inherent structure to prevent coupling between the RF pogo pins. While this pogo-pin array may not pose a problem in single-port situations or at low frequencies below megahertz, there is a structural limitation that cannot effectively block RF leakage between signal pins in the high-frequency band. Furthermore, the height of the pogo pin, which is 2.8 mm and satisfies half the wavelength at 28 GHz in contact conditions, can lead to the dielectric socket itself becoming a resonator within the operating frequency band due to leakage signals from coupled power. This dielectric resonator structure can also contribute to the coupling component. Unlike a single-port environment in previous studies where pins around the signal are absent or grounded, additional issues may arise in a multi-port array environment. Therefore, solving the leakage problem is essential in high-frequency bands.

## 3. Design of Multi-Port Probe Card with Metallic Pogo-Pin Socket

To address these issues, the pogo-pin probe card shown in Figure 1 is proposed. The designed pogo-pin probe card has been significantly reduced in size compared to the previous version. As shown in Figure 1, the length of the PCB assembled on the probe card is 21 mm, which is less than 1/3 of the length used in the previous study. This structure offers the advantage of having a shorter PCB line length, approximately 5.73 λ at 50 GHz. A compact probe card PCB structure can achieve low insertion loss in the through line connected with pogo pins. This also facilitates analysis by allowing easy confirmation of coupling loss due to the low insertion loss characteristics. The designed probe card PCB utilizes a TLY-5 substrate (dielectric constant 2.2), and a low insertion loss performance of −1.46 dB at 50 GHz for the same line length is confirmed through measurement. To implement such a compact probe card, the pogo-pin socket integrated into the probe card is also miniaturized. The detailed structure of the pogo-pin socket shown in Figure 1 is depicted in Figure 2a. The proposed pogo-pin socket is designed with a short length of 10 mm. The central part of the socket is the circuit area used for measuring the circuit, and a pogo-pin array is positioned inside this area based on the ball map of the RF chip. Apart from the circuit area, the remaining part serves as the holder structure and is designed to be compact, unlike typical pogo-pin sockets used previously. The holder part of the probe card is designed to be the smallest size possible for assembly. A robust integration of the PCB and the pogo-pin socket is essential for the probe card configuration, and this integration is facilitated by the screw hole (white ellipses on both sides of the socket) shown in Figure 1. Through prototype fabrication, it is confirmed that fabricating the holder smaller than this size makes screw assembly challenging, thus hindering the construction of the probe card.

The pogo-pin socket structure, made of dielectric material, is unable to eliminate coupled power between closely spaced pogo pins because it lacks the capability to block RF signal leakage in a dielectric structure. In previous studies, ground pins were placed close together to prevent coupling [22]. However, as the frequency increases, this approach leads to impedance mismatch, making it challenging to use in the millimeter-wave band. Fabrication also becomes challenging because the ground pins must be placed in close proximity to the signal pogo pins. To solve the coupling problem of the pogo-pin socket in the high-frequency band and enable the design of a fabrication feasible pogo-pin socket, it is necessary to consider changing the socket material. Figure 2a displays a photograph of the pogo-pin socket made with metal material. Complete isolation between pogo pins is achieved by locating a metal wall between the pogo pins. The designed metallic pogo-pin socket shown in Figure 2a consists of two parts: the main body part where the pogo pins are positioned and a spacer that secures the pogo pins. Unlike previous dielectric pogo-pin sockets, only the main body section of the pogo-pin socket is replaced with metal, while the material of the spacer remains dielectric. The spacer material remains unchanged because replacing it with metal could cause an RF short between the signal pin and ground pin through the spacer. As shown in Figure 2a, the central portion of the pogo-pin socket is the circuit area, with a pogo-pin array (marked by the red dashed line) positioned within this area. The remaining part functions as a socket holder for assembly with the probe card PCB. The designed metallic pogo-pin socket features a pogo-pin array consisting of 4 signal pogo pins and 12 ground pins. Each of the four signal pins is connected to port 1 (input port), port 2 (through port), port 3 (coupled port), and port 4 (isolated port) of the probe card PCB, respectively, forming a four-port system. Although the actual pogo-pin socket used for measuring a multi-port RF system in the circuit area may involve more than 100 pogo pins, this study focuses on a prototype design with four ports to analyze the insertion performance and coupled components of the proposed metallic pogo-pin socket.

Figure 2b provides a detailed view of the pogo-pin array section of the metallic pogo-pin socket. To minimize leakage signals that may occur within the dielectric spacer, the pogo-pin array is structured with four ground pins surrounding each signal pogo pin. The first and third columns consist of the ground pin array, while the second column is dedicated to the RF signal pin array. The structure of the signal pin is designed in a coaxial-type configuration, where a signal pogo pin is placed inside a non-shorting air gap to prevent RF shorting. The diameter of the designed non-shorting air gap is 0.34 mm, which is intended to satisfy the 50 Ω characteristic impedance of the RF pogo pin. The holes placed in the ground pins are designed to match the diameter of the pins, ensuring that the metallic socket itself is connected to the ground. Within the spacer, the distance between the signal pin and the ground pin is maintained at 0.35 mm to achieve a 50 Ω impedance, and the distance between the RF pogo pins is set close to 0.522 mm. The dielectric spacer used in the pogo-pin socket is designed to be 0.5 mm thick to maintain strength and secure the pogo pins using the structure of pogo pins with different barrel and plunger diameters. The metal wall (green structure in Figure 2b) provides a complete isolation structure between RF pogo pins, ensuring that issues such as dielectric resonators and coupling line problems associated with dielectric sockets are eliminated.

## 4. Analysis of Coupling Problem in Dielectric Pogo-Pin Socket and Fabrication of Prototype Metallic Socket Probe Card

The metallic pogo-pin socket proposed in Section 3 features a metal wall between the pogo pins, allowing each pogo pin to maintain a 50 Ω characteristic impedance. Therefore, when performing coupling modeling, the 50 Ω characteristics are maintained in both odd mode and even mode. In contrast, the dielectric pogo-pin socket cannot maintain a 50 Ω characteristic impedance in a closely spaced pogo-pin environment due to the lack of a pin-to-pin isolation structure. Figure 3 shows the modeling of a coupled line in a dielectric pogo-pin socket. As depicted, the coupled lines are closely spaced, with a distance between RF pogo pins at 28 GHz corresponding to 0.09 λ, facilitating significant signal crossover to adjacent pogo pins. Moreover, the pogo-pin structure is not solely a coupled-line structure; pogo-pin probe cards encompass complex elements such as pad structures for PCB connection and the dielectric resonator inherent to the dielectric socket itself. Therefore, circuit model analysis is conducted using the Advanced Design System (ADS) program. It is determined that the odd-mode impedance should be around 30 Ω, and the even-mode impedance should be approximately 110 Ω through the coupled power simulation of the entire structure of the pogo-pin probe card. Additionally, due to the narrow bandwidth characteristic compared to a typical coupled line, structures such as dielectrics and pads are expected to contribute to capacitance. Based on the ADS calculation, a capacitance of approximately 30 fF is predicted. Consequently, coupling issues become challenging to mitigate when using a dielectric socket at high frequencies. This is attributed to the dielectric material potentially acting as a conduit for RF signals, making complete leakage removal impossible even with the presence of ground pins.

A prototype probe card designed to verify the issues in a multi-port environment up to 50 GHz and test the solutions provided by proposed metallic pogo-pin sockets has been fabricated, as shown in Figure 4a. The final design includes a metallic pogo-pin socket with a 4-port configuration (2-by-2 port), assembled with a probe card PCB and 2.4 mm RF connectors. The metallic socket is made of brass material and is designed with a 0.34 mm RF signal pin hole and 0.16 mm ground pin hole. The reason for fabricating the ground pin hole to be slightly larger than the barrel diameter (0.15 mm) is to account for potential manufacturing errors. The designed dielectric spacer is made of polyamide–imide (PAI) material with a dielectric constant of 3.9 and low dielectric loss characteristics. The height of the probe card, including the thickness of the socket main body and the spacer, is 2.8 mm. The pogo pins used in the socket are GN1087CR-DGPC models from Leeno Industrial Inc., with a barrel diameter of 0.15 mm, and the pogo pins are arranged at intervals of 0.35 mm. The PCB used for the probe card, which transmits the RF signal to the pogo pins, is fabricated using a 0.127 mm thick TLY-5 substrate (dielectric constant 2.2), known for its low dielectric loss characteristics even at high frequencies. A 2.4 mm OS-50 (Hirose H24LRSR2) connector is attached to the end of the PCB line to ensure stable transmission performance up to 50 GHz. By securing the connector with a screw, the impact of contact loss can be minimized. A dielectric socket of the same structure is also fabricated for comparison purposes. The material of this socket is PAI, with the same material as the spacer. The only difference between the fabricated dielectric socket and the metallic socket lies in the material used for the main body part. The dielectric pogo-pin socket is designed with the same width of 10 mm and the same spacing as the metallic socket. An RF signal pin surrounded by four ground pins is used in the dielectric socket to minimize the coupling component to adjacent pogo pins. The pogo pins used in the dielectric socket are the same model as those used in the metallic socket.

The measurement setup, as shown in Figure 4b, is configured to measure the prototype probe card. The measurements are conducted using a 50 GHz vector network analyzer (VNA, E8364B) to assess insertion loss and coupling loss. To measure these parameters effectively, a 4-port measurement setup is required. However, due to limitations in the VNA structure, the non-measurement port is connected with a 50 Ω termination, as depicted in Figure 4b. The measurement process involves using a short through-test line located at the bottom of the probe card. To ensure accurate performance measurement of the pogo-pin probe card, the VNA and the probe card are connected using a low-loss OS 50 cable (TCF219). Calibration is performed using a 2.4 mm 85056D calibration kit at the connector position to facilitate precise measurement of the probe card’s performance.

## 5. Verification and Measurement of Multi-Port Probe Card with Metallic Socket and Test Board

Since the designed pogo-pin probe card is intended for verifying RF circuits, it is necessary to validate it using test circuit measurements. The test circuit used in this study should be simple and should not affect the performance of the probe card. Therefore, the test circuit is a basic through-line configuration consisting of two lines to assess the performance in a four-port system. The structure of the through line used for the measurement is shown in Figure 5a. To prevent coupling issues in the test through line, it has been designed with a spacing of 2.3 mm between the lines. To maintain a characteristic impedance of 50 Ω, the test line is also configured with a width of 0.365 mm, matching the width of the probe card PCB line. However, using this width for the test line can be challenging to fabricate in a standard microstrip line configuration on a PCB. This is due to the narrow spacing between the ground pins and the relatively wide line width, which results in the ground pad being positioned very close to the line. When the ground pad is positioned close to the line, it becomes challenging to maintain a characteristic impedance of 50 Ω, which affects the accuracy of checking the probe card’s performance. Consequently, the test circuit is designed in an inverted microstrip configuration, as depicted in Figure 5a. As shown in the left photograph in Figure 5a, the RF pad is located on the top side, while the through line is fabricated on the bottom side. Since the line is beneath the PCB, measurements are conducted using a polystyrene material chuck with a dielectric constant close to 1 and minimal dielectric loss.

Both the metallic pogo-pin socket and the dielectric socket are subjected to measurements using the fabricated test board. Additionally, the High-Frequency Structure Simulator (HFSS) is employed to predict the performance of the probe card structure. Figure 5b–d show the simulation results using HFSS and the measurement results of the probe card with the metallic pogo-pin socket and the dielectric socket along with the test line. Figure 5b shows the insertion loss of the probe card in a multi-port environment. In Figure 5b, the red solid line represents the measurement result using a metallic socket, while the dashed line indicates the simulation result. The measurement and simulation results closely align, demonstrating that using a metallic socket significantly reduces insertion loss by eliminating the cause of coupled power. As a result, it can be confirmed that there are no issues related to coupling, demonstrating an insertion loss performance of −4.28 dB at 50 GHz. Therefore, it is possible to deliver a power of −2.14 dB or more to the RF chip at 50 GHz. Additionally, in a single-port environment, the −3.63 dB of insertion loss result up to 50 GHz demonstrates similar performance to that of the multi-port pogo-pin socket.

On the other hand, when simulating and measuring with a dielectric socket, Figure 5b reveals a coupling issue at specific frequencies. The blue line in Figure 5b represents the results using a dielectric socket. Although the insertion loss of −3.91 dB at 50 GHz is similar to that of the metallic socket, problems arise at certain frequencies. In contrast to the results with the metallic socket, the dielectric socket exhibits significant insertion loss due to coupled loss. Specifically, the transmission performance decreases to −5.94 dB at 12.7 GHz. The spurious coupling frequency of 12.7 GHz generated in the dielectric socket is determined by the height of the pogo pin that constitutes the coupled line. The length of the pogo-pin product used in the prototype probe card is 2.8 mm, which is similar to the length of 0.25 wavelength inside the dielectric (PAI material) socket. Therefore, the coupled line resonance problem occurs at frequencies close to that length. This performance deterioration is evident both in the simulation and measurement results. The simulation predicts a decline in performance due to coupled power, even at higher frequencies. However, at these frequencies, the inherent loss is significant, making it challenging to confirm the trend definitively. However, in the 13 GHz band, the problem caused by coupling can be verified. The insertion loss results highlight the coupling issue inherent in the dielectric socket. 

Figure 5c shows the measured coupled power associated with the coupling port 3. With the metallic pogo-pin socket, it is possible to maintain a low coupled power performance of −15.67 dB or less across all frequency bands. However, with the dielectric socket, the coupled power increases significantly to −7.79 dB at a frequency of 12.7 GHz. This increase in coupled power is observed in both the measurement and simulation, indicating higher coupling levels at 12.7 GHz. Completely eliminating leakage through the pogo-pin socket made of dielectric material, even with ground pins surrounding the signal pogo pin, proves challenging. Despite the lower coupled power than predicted by simulation because of dielectric loss, it remains a concern when compared to the performance of the metallic socket. In the high-frequency band, the simulation and measurement plots exhibit similar trends, but the coupled loss appears lower in the measurement due to the dielectric loss of the socket and PCB. At 12.7 GHz, the difference between the coupled power and the transmission performance is only 2 dB, which may lead to RF signal mixing issues. This mixing can result in significant signal distortion and increased noise. Furthermore, Figure 5d shows the return loss, highlighting the occurrence of high reflection loss in the dielectric socket at frequencies where coupling issues arise. Based on the measurement results, changing the socket material to metal confirms the potential relaxation of coupling components. Table 1 presents the performance comparison of various pogo-pin probe cards. The probe card proposed in this paper achieved a lower insertion loss performance with a narrow pitch environment up to 50 GHz than previous studies and low coupled power performance up to 50 GHz.

Compared to the simulation result, the measurement result exhibits additional ripples. Two additional measurements are conducted to analyze the cause of the ripple loss. Predicting that the cause of the ripple is due to the contact section, we analyze the contact section between the connector and the PCB line and the contact section between the PCB and socket connection. The two added measurement results are shown in Figure 6. Figure 6a is a photograph of the PCB line without a socket. Figure 6b is a result of measuring the loss of the only PCB line without a socket, including connector–PCB contact loss. Figure 6c is the RF short measurement result. It is measured by contacting the pogo pin of the probe card with the metal surface. The measurement result is carried out to check whether assembly loss occurred between the PCB and the socket. Based on the two additional measurement results for the analysis, it can be confirmed that ripples occur at high frequencies due to various connection sections, unlike low frequencies. Unlike the simulation in which linear results are shown, an OS-50 connector is used for actual measurement assembly, but it is challenging to achieve complete 50 Ω impedance when it contacts the PCB line, leading to a slight reflection that causes standing waves. Insertion loss is high at certain frequencies (24 GHz, 30 GHz, and 39 GHz) in the high-frequency band by ripples. In particular, the ripple is high at the socket and PCB connection part. The cause of this loss occurs due to the difference in design from the actual fabricated PCB. The PCB and socket are contacted through a pad. There is no problem in the design and simulation because this pad is designed so that there is no loss. However, there is a fabricating tolerance in the actual fabrication, causing a problem. Due to the narrow gap between pogo pins, the pad itself is a very small structure, so it is not complete to implement with PCB manufacturing precision. Unlike the design, it is judged that contact loss occurred due to the deformed pad structure. In order to minimize the loss due to such a connection, a broadband matching structure between PCB and connector is considered to be necessary. In addition, it is necessary to minimize the loss through accurate via pad structure production.

Additionally, PCB circuit measurements consisting of multiple ports are conducted to verify that the designed pogo-pin metallic socket can accurately measure the properties of the multi-port chip. The fabricated multi-port passive circuit is a simple T-junction line, as shown in Figure 7a, and consists of three ports. It is designed with an inverted microstrip configuration similar to the through line used for the probe card test. It is designed to ensure line spacing, as shown in Figure 7a, to avoid coupling problems that might occur in the PCB circuit itself. The transmission performance of the T-junction circuit is designed to deliver approximately the same power as −3.7 dB to port 2 and port 3, and this performance is confirmed through HFSS simulation. The T-junction line is implemented on the same TLY-5 substrate used for the through line. The insertion loss results of measuring the designed T-junction circuit with a metallic socket probe card (solid line) and a dielectric probe card (dashed line) are shown in Figure 7b. When a metallic socket probe card is used, the probe card insertion loss increases linearly with frequency. The transmission performance is −5.1 dB at 20 GHz frequency to each port, and no coupling problems occur, as observed with the through line.

On the other hand, when the T-junction line is measured using a dielectric socket, the coupling problem of the dielectric socket is reflected in the measurement results. The RF signal distortion caused by coupling leads to an insertion loss decrease to −15.97 dB at 11.6 GHz, and a problem of −9.45 dB at 13.3 GHz is observed. Overall, the insertion loss deteriorates due to the coupling problem when a dielectric socket is used. When multi-port circuits like the T-junction line are tested with a dielectric pogo-pin socket, the probe card structure can affect the test circuit’s performance or lead to issues such as frequency shifts. Therefore, even if calibration is performed at the end of the pogo-pin probe card, these problems may persist. Therefore, using a dielectric socket for a millimeter-wave band probe card that needs to be measured up to several tens of GHz poses a risk. However, since the metallic socket can solve this problem, it can be considered more suitable for 5G band multi-port RF circuit measurement.

## 6. Conclusions

A structure for a pogo-pin probe card that incorporates a metallic socket to address coupling issues in a multi-port environment is proposed. Unlike single-port setups, multi-port configurations can encounter coupling problems. The designed metallic socket features a complete isolation structure between signal pogo pins, effectively preventing coupling and eliminating the possibility of the socket itself acting as a resonator, which could affect coupling components in a multi-port environment. The metallic socket offers the advantage of reducing coupled power but presents a challenge in terms of fabricating a design that securely holds pogo pins without causing RF short. To solve this problem, dielectric spacers are placed on both sides of the metallic sockets to prevent RF shorting and to securely fix the pins. A prototype of the metallic pogo-pin socket probe card, including the dielectric spacer, is fabricated, and the performance of the metallic socket is verified through measurements in this study. When comparing the dielectric socket with the same structure as the designed metallic socket, a significant coupling issue at specific frequencies is observed with the dielectric socket. However, measurements demonstrate that the designed metallic socket effectively eliminates this coupling problem. Without coupled loss, the insertion loss remains linear up to 50 GHz, with the maximum insertion loss kept below 5 dB. Furthermore, the coupled power is maintained below −15 dB, which prevents distortion between RF signals even in multi-port setups. The compact size of the metallic socket also reduces RF signal insertion loss at 50 GHz compared to previous pogo-pin sockets. In the chip testing results, measurements show a minimal loss of only 0.7 dB at 28 GHz. The proposed probe card enables the practical use of test facilities in the 5G band by fundamentally eliminating issues that may arise at high frequencies and in multi-port setups, such as millimeter-wave bands. This technology is valuable for measuring multi-port array systems and MIMO systems, which are predominantly designed for millimeter wave band frequencies. It has wide applicability in the commercial chip verification industry for 5G bands.

## Figures and Tables

**Figure 1 sensors-24-03745-f001:**
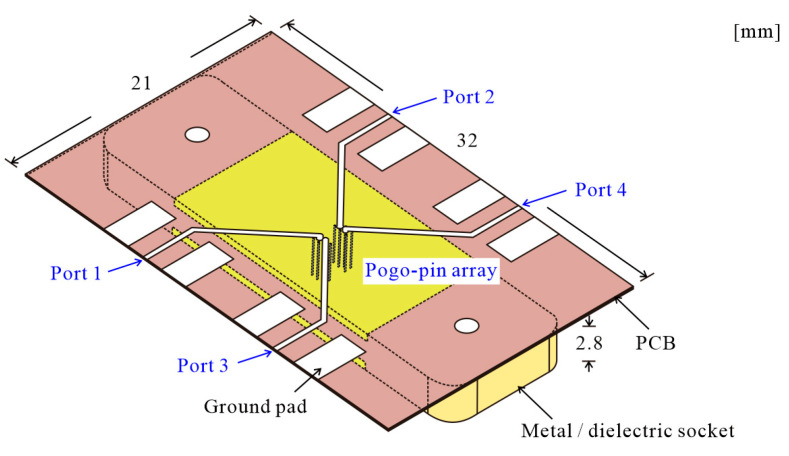
Sketch of four-port probe card system.

**Figure 2 sensors-24-03745-f002:**
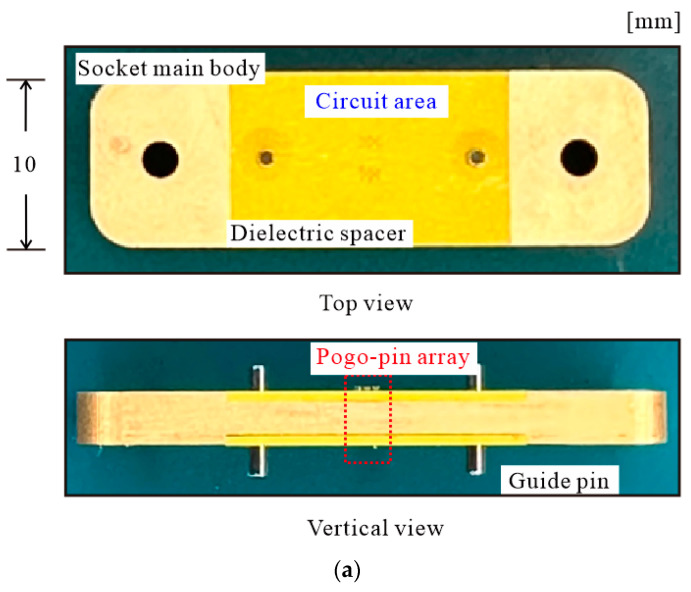
The pogo-pin socket: (**a**) photograph and (**b**) top- and side-view drawings.

**Figure 3 sensors-24-03745-f003:**
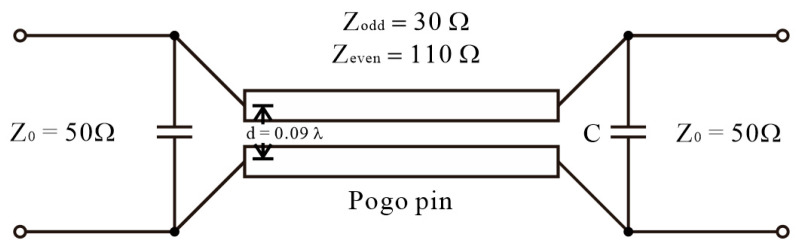
Coupled-line circuit model for neighboring pogo pins placed in the dielectric socket.

**Figure 4 sensors-24-03745-f004:**
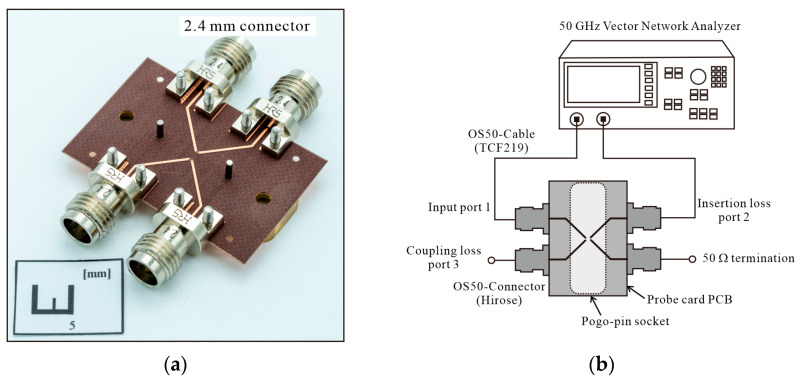
(**a**) Photograph of complete probe card system and (**b**) small-signal measurement setup.

**Figure 5 sensors-24-03745-f005:**
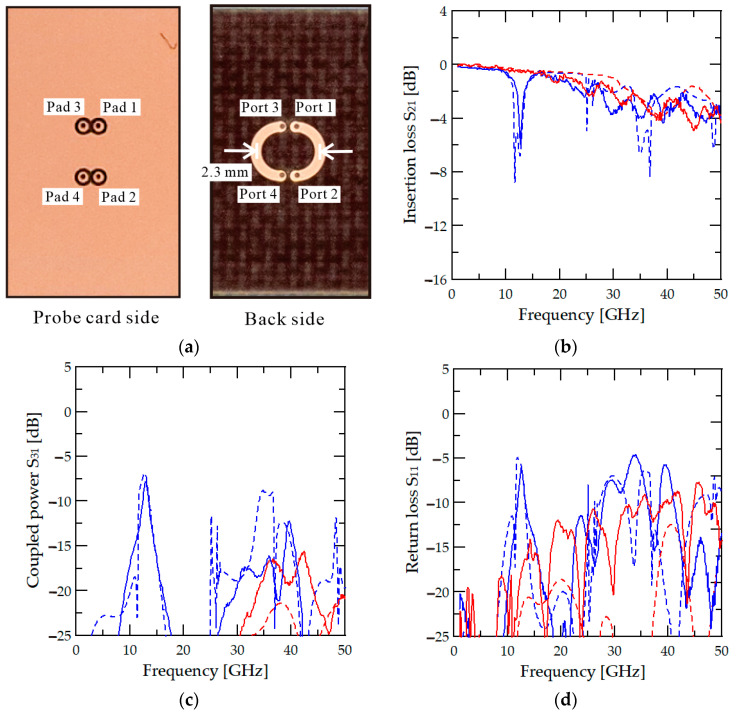
(**a**) Photograph of the through-line circuit and measured (**b**) S21, (**c**) S11, and (**d**) S31 for dielectric (blue) and for metal (red) sockets. Measured results are shown with solid lines, and simulation results are represented with dashed lines.

**Figure 6 sensors-24-03745-f006:**
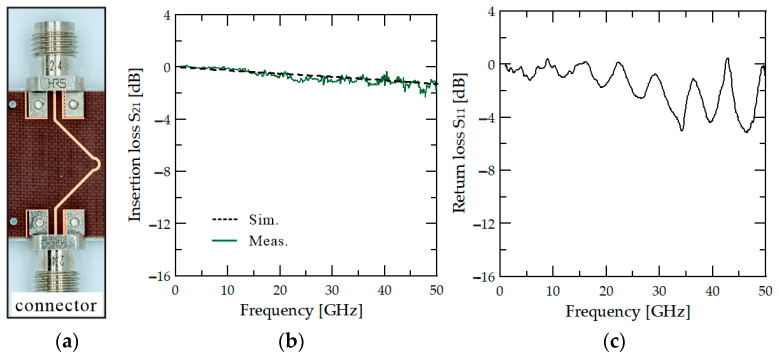
(**a**) The photograph of simple through line and measured (**b**) insertion loss of through line, and (**c**) return loss of short connection. Measured results are shown in solid and simulation in dashed lines.

**Figure 7 sensors-24-03745-f007:**
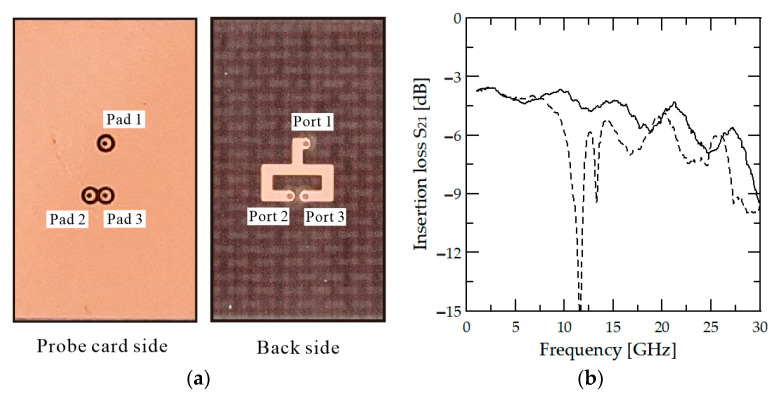
(**a**) The photograph of simple T-junction circuit and (**b**) the insertion losses for metallic socket (solid) and dielectric socket (dashed).

**Table 1 sensors-24-03745-t001:** Performances of multiple pogo-pin RF probes.

Ref.	Max. Frequency	Port	Type	Pitch	Insertion Loss	Coupled Power (max.)
[13]	20 GHz	Single	Pogo pin	0.8 mm	3 dB @20 GHz	-
[14]	10 GHz	Multi	Pogo pin	1.27 mm	8 dB @10 GHz	−20 dB @10 GHz
[16]	40 GHz	Single	Pogo pin	2.54 mm	6 dB @28 GHz	-
[17]	40 GHz	Single	Pogo pin	0.8 mm	0.5 dB @40 GHz	-
[19]	20 GHz	Single	Pogo pin	0.616 mm	12 dB @20 GHz	-
[20]	25 GHz	Single	Pogo pin	0.35 mm	0.5 dB @25 GHz	-
[21]	50 GHz	Single	Pogo pin	0.35 mm	3.1 dB @50 GHz	-
[22]	10 GHz	Multi	Pogo pin	0.177 mm	1.93 dB @10 GHz (Sim.)	−26.3 dB @10 GHz (Sim.)
[23]	10 GHz	Multi (2 × 2)	Pogo pin	1.8 mm	-	−20 dB @10 GHz
[25]	20 GHz	Multi	Pogo pin	0.177 mm	2.37 dB @10 GHz (Sim.)	−26.3 dB @20 GHz (Sim.)
This work	50 GHz	Multi (2 × 2)	Pogo pin	0.35 mm	2.1 dB @50 GHz	−15.67 dB @42.5 GHz

## Data Availability

The data presented in this study are available on request from the corresponding author.

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
