# Peer review of "50 GHz Four-Port Coupling-Reduced Probe Card Utilizing Pogo Pins Housed in Custom Metallic Socket"

_sensors, 2024, doi:10.3390/s24123745_

Round 1

Reviewer 1 Report

Comments and Suggestions for Authors

The proposed probe board structure with immersed pins in a multiport environment is of interest. Indeed multiport configurations can bring additional coupling problems as shown in the paper. However, there are questions about setting up measurements:

Coax connector used to connect the microstrip line brings additional reflection and insertion loss. To complete the measurement procedure, it would be necessary to present the results of the line loaded on the connectors in order to estimate the mismatching and losses in the PCB lines and coax to microstrip transition. The best option is to provide a calibration at the pogo pins plane in order to de-embed the adapters and PCB lines.

Secondly, there are not enough comments about the increased losses for the thru-line circuit in the region of 24 GHz, 30 GHz and 39 GHz (metal sockets, Fig.5). Reflection loss and coupling are acceptable, but losses are more significant compared to other frequencies.

Author Response

Review 1

The proposed probe board structure with immersed pins in a multiport environment is of interest. Indeed multiport configurations can bring additional coupling problems as shown in the paper. However, there are questions about setting up measurements:

(1) Coax connector used to connect the microstrip line brings additional reflection and insertion loss. To complete the measurement procedure, it would be necessary to present the results of the line loaded on the connectors in order to estimate the mismatching and losses in the PCB lines and coax to microstrip transition. The best option is to provide a calibration at the pogo pins plane in order to de-embed the adapters and PCB lines.

1. Thank you for your important review comments. De-embed is the correct analysis to check the performance of the metallic socket only. However, we intend that the probe card for the on-wafer measurement in millimeter wave is a structure that includes both a metallic socket and a PCB for signal transmission. Our research goal is to deliver the RF signal power entered into the probe card to the integrated chip without any coupling loss, resonance loss, and material loss, so the loss generated from the PCB is also an important factor. We calibrated at the PCB input part to understand the problems that occur throughout the probe card including PCB structural performance. In addition, we designed it to miniaturize both socket and PCB to minimize the loss of the entire probe card.

We checked the opinion you gave us and decided that the analysis of the measurement result was insufficient, so we proceeded with additional measurement for analysis and proceeded with more detailed analysis. The additional measurement for analysis was added as Figure 6. Further analysis helped to determine the cause of the loss clearly. Thank you for giving me the opportunity to correct it with your accurate review.

(2) There are not enough comments about the increased losses for the thru-line circuit in the region of 24 GHz, 30 GHz and 39 GHz (metal sockets, Fig.5). Reflection loss and coupling are acceptable, but losses are more significant compared to other frequencies.

2. The added Figure 6(b) was a result of measuring the loss of the only PCB line without socket includes connector and PCB contact loss. Figure 6(c) was the RF short measurement result. It was measured by contacting the pogo pin with the metal surface. The measurement was carried out to check whether assembly loss occurred between the PCB and the socket.

Based on the two additional measurement results for the analyze, it can be confirmed that ripples occur at high frequencies due to various connection sections, unlike low frequencies. In simulation, only linear results are shown, but in measurement, an issue due to connection occurs. Insertion loss is high at certain frequencies (24 GHz, 30 GHz, 39 GHz) in the high frequency band by ripples.

In particular, the ripple is high at the socket and PCB connection part. The cause of this loss occurs due to the difference in design from the actual fabricated board. The PCB and socket are contacted through via pad. There is no problem in the design because this pad is designed so that there is no loss, but there was a fabricating tolerance in the actual fabrication, causing a problem. Due to the narrow gap between pogo pins, the pad itself is a very small structure, so it was not complete to implement with PCB manufacturing precision. Unlike the design, it is judged that contact loss occurred due to the deformed pad structure.

In order to minimize the loss due to such a connection, a broadband matching structure between PCB and connector is considered to be necessary. In addition, it is necessary to minimize the loss through accurate via pad structure production.

We organized the above and added it to the manuscript as well.

Reviewer 2 Report

Comments and Suggestions for Authors

The initial part of the introduction (up to line 35) seems to be a bit confusing, generally referring to different fields and without a concrete focus on on-wafer multiport systems at mm-wave. For example, see systems like C. Schulze et al., "A VNA-Based Wideband Measurement System for Large-Signal Characterization of Multiport Circuits," in IEEE Transactions on Microwave Theory and Techniques, vol. 72, no. 1, pp. 638-647, Jan. 2024, which could be one of the main prospective application cases for the proposed pin device. Also, the authors should provide a more comprehensive comparison with the most common solution (i.e., multiprobe cards and/or dual/differential coplanar probes).

The second part of the introduction (up to line 101) provides relevant and interesting references on the solutions that could be found in the literature, especially commenting on high-frequency behavior. However, given the many mentioned references, it would be useful if the authors could include some summary plots/graphical diagrams and/or a table highlighting the main characteristics (frequency, insertion loss, dimensions, etc.) to enable an immediate comparison.

From line 102, the authors should better highlight the novelty of their solution. As written, it seems that the only novelty is geometrical/mechanical (new metallic socket). Is that the case? Please improve the commentary on the differences with respect to the state of the art.

A large part of Section 2 (up to line 153) is dedicated to describing the problems of a previous design based on a dielectric socket. This part is very well written and must be part of the article; however, I suggest dividing Section 2 into two subsections, where the first subsection is related to the trade-offs of this previous design, whereas the second subsection (from line 154 till the end of Section 2) actually describes the new design. Another alternative would be to split Section 2 into two sections.

In the description of the measurement setup, the authors mention that they calibrate up to the connector plane. However, in order to assess the performance of the pin socket only, one should de-embed the effect of the coplanar lines of the PCB. This doubt is also related to what the authors mention in line 329 concerning the connector: "it is challenging to achieve complete 50-ohm impedance when it contacts the PCB line, leading to a slight reflection that causes standing waves." Could the authors clarify these aspects?

Concerning the results in Figs. 5 and 6: the insertion loss becomes quite high at high frequencies. Could the authors suggest some technological improvements in this sense? Also, the spurious coupling at 12.7 GHz should be better investigated, since only a general commentary and no clear explanation is provided.

Author Response

Review 2

(1) The initial part of the introduction (up to line 35) seems to be a bit confusing, generally referring to different fields and without a concrete focus on on-wafer multiport systems at mm-wave. For example, see systems like C. Schulze et al., "A VNA-Based Wideband Measurement System for Large-Signal Characterization of Multiport Circuits," in IEEE Transactions on Microwave Theory and Techniques, vol. 72, no. 1, pp. 638-647, Jan. 2024, which could be one of the main prospective application cases for the proposed pin device. Also, the authors should provide a more comprehensive comparison with the most common solution (i.e., multiprobe cards and/or dual/differential coplanar probes).

1. As you pointed out, the beginning part of the introduction (up to line 35) was ambiguous, so we reinforced and modified the content. We revised the contents of this part with a focus on the multi-port probe card that can measure multi-port RF chips. In addition, a commercialized probe card reference was added to further explain the existing problems claimed in the paper.

Thank you for your detailed review and thank you for your good reference.

(2) The second part of the introduction (up to line 101) provides relevant and interesting references on the solutions that could be found in the literature, especially commenting on high-frequency behavior. However, given the many mentioned references, it would be useful if the authors could include some summary plots/graphical diagrams and/or a table highlighting the main characteristics (frequency, insertion loss, dimensions, etc.) to enable an immediate comparison.

2. As you advised, we added table 1 in the manuscript.

Table 1. Performances of multiple-pogo pin RF probes.

Ref.

Max. Frequency

Port

Type

Pitch

Insertion Loss

Coupled Power (max.)

[13]

20 GHz

Single

Pogo pin

0.8 mm

3 dB @20 GHz

-

[14]

10 GHz

Multi

Pogo pin

1.27 mm

8 dB @10 GHz

−20 dB @10 GHz

[16]

40 GHz

Single

Pogo pin

2.54 mm

6 dB @28 GHz

-

[17]

40 GHz

Single

Pogo pin

0.8 mm

0.5 dB @40 GHz

-

[19]

20 GHz

Single

Pogo pin

0.616 mm

12 dB @20 GHz

-

[20]

25 GHz

Single

Pogo pin

0.35 mm

0.5 dB @25 GHz

-

[21]

50 GHz

Single

Pogo pin

0.35 mm

3.1 dB @50 GHz

-

[22]

10 GHz

Multi

Pogo pin

0.177 mm

1.93 dB @10 GHz (Sim.)

−26.3 dB @10 GHz (Sim.)

[23]

10 GHz

Multi (2x2)

Pogo pin

1.8 mm

-

−20 dB @10 GHz

[25]

20 GHz

Multi

Pogo pin

0.177 mm

2.37 dB @10 GHz (Sim.)

−26.3 dB @20 GHz (Sim.)

This work

50 GHz

Multi (2x2)

Pogo pin

0.35 mm

2.1 dB @50 GHz

−15.67 dB @42.5 GHz

(3) From line 102, the authors should better highlight the novelty of their solution. As written, it seems that the only novelty is geometrical/mechanical (new metallic socket). Is that the case? Please improve the commentary on the differences with respect to the state of the art.

3. As you advised, we judged that the explanation was very poor, so we added explanations for each section and explained the structure we proposed in more detail.

The novelty that we're advocating is a socket structure using a metal structure. The change in material may seem too simple and nothing special. But we intended to verify that even this simple thought transition would eliminate the problem of high frequency bands. In this paper, we also propose a structure that can solve RF short and pin fixing problems that can be caused by using metal materials, not just explaining the possibility of change to metal materials. We also tried to identify and solve the problems that occur in the fabrication of the entire probe card system at high frequencies.

(4) A large part of Section 2 (up to line 153) is dedicated to describing the problems of a previous design based on a dielectric socket. This part is very well written and must be part of the article; however, I suggest dividing Section 2 into two subsections, where the first subsection is related to the trade-offs of this previous design, whereas the second subsection (from line 154 till the end of Section 2) actually describes the new design. Another alternative would be to split Section 2 into two sections.

4. Thanks for very helpful suggestions. As you advised, we modified to split Section 2 into two sections.

(5) In the description of the measurement setup, the authors mention that they calibrate up to the connector plane. However, in order to assess the performance of the pin socket only, one should de-embed the effect of the coplanar lines of the PCB. This doubt is also related to what the authors mention in line 329 concerning the connector: "it is challenging to achieve complete 50-ohm impedance when it contacts the PCB line, leading to a slight reflection that causes standing waves." Could the authors clarify these aspects?

5. Thank you for your important review comments. De-embed is the correct analysis to check the performance of the metallic socket only. However, we intend that the probe card for the on-wafer measurement in millimeter wave is a structure that includes both a metallic socket and a PCB for signal transmission. Our research goal is to deliver the RF signal power entered into the probe card to the integrated chip without any coupling loss, resonance loss, and material loss, so the loss generated from the PCB is also an important factor. We calibrated at the PCB input part to understand the problems that occur throughout the probe card including PCB structural performance. In addition, we designed it to miniaturize both socket and PCB to minimize the loss of the entire probe card.

We checked the opinion you gave us and decided that the analysis of the measurement result was insufficient, so we proceeded with additional measurement for analysis and proceeded with more detailed analysis. The additional measurement for analysis was added as Figure 6. Further analysis helped to determine the cause of the loss clearly. Thank you for giving me the opportunity to correct it with your accurate review.

(6) Concerning the results in Figs. 5 and 6: the insertion loss becomes quite high at high frequencies. Could the authors suggest some technological improvements in this sense? Also, the spurious coupling at 12.7 GHz should be better investigated, since only a general commentary and no clear explanation is provided.

6. The added Figure 6(b) was a result of measuring the loss of the only PCB line without socket includes connector and PCB contact loss. Figure 6(c) was the RF short measurement result. It was measured by contacting the pogo pin with the metal surface. The measurement was carried out to check whether assembly loss occurred between the PCB and the socket.

Based on the two additional measurement results for the analyze, it can be confirmed that ripples occur at high frequencies due to various connection sections, unlike low frequencies. In simulation, only linear results are shown, but in measurement, an issue due to connection occurs. Insertion loss is high at certain frequencies (24 GHz, 30 GHz, 39 GHz) in the high frequency band by ripples.

In particular, the ripple is high at the socket and PCB connection part. The cause of this loss occurs due to the difference in design from the actual fabricated board. The PCB and socket are contacted through via pad. There is no problem in the design because this pad is designed so that there is no loss, but there was a fabricating tolerance in the actual fabrication, causing a problem. Due to the narrow gap between pogo pins, the pad itself is a very small structure, so it was not complete to implement with PCB manufacturing precision. Unlike the design, it is judged that contact loss occurred due to the deformed pad structure.

In order to minimize the loss due to such a connection, a broadband matching structure between PCB and connector is considered to be necessary. In addition, it is necessary to minimize the loss through accurate via pad structure production.

The spurious coupling frequency of 12.7 GHz generated in the dielectric socket is determined by the length of the pogo pin that constitutes the coupled line. The length of the pogo pin product used in design and actual manufacture is 2.8 mm, which is similar to the length of 0.25 wavelength inside the dielectric(PAI material) socket. Therefore, the coupled line resonance problem occurs at frequencies close to that length.

We organized the above and added it to the manuscript as well.

Round 2

Reviewer 2 Report

Comments and Suggestions for Authors

Thanks for addressing the comments.